# Dynamics of CO photooxidation to CO$_2$ on rutile (110)
Helena Gleissner [1,2,3], Michael Wagstaffe [1,10], Lukas Wenthaus [4], Adrian Domínguez-Castro [5], Verena Gupta [5], Simon Chung [1], Steffen Palutke [4,11], Siarhei Dziarzhytski [4], Dmytro Kutnyakhov [4], Michael Heber [4,11], Günter Brenner [4], Harald Redlin[4], Federico Pressacco [4], Adriel Domínguez Garcia[5,6,7], Thomas Frauenheim[8,9], Heshmat Noei [1,2] ✉ & Andreas Stierle [1,2,3] ✉

Free-electron lasers (FELs) enable the study of the ultrafast dynamics of photocatalytic reactions by time-resolved X-ray photoelectron spectroscopy (tr-XPS) with femtosecond time resolution. In an optical pump - soft X-ray probe photoemission experiment conducted at the free-electron laser in Hamburg (FLASH), we observed the ultrafast oxidation of CO to CO$_2$ on rutile TiO$_2$(110) by monitoring the O 1s core level region. Within 800± 250 fs after laser excitation, CO$_2$ as a product of the photooxidation of CO is detected. Based on density functional theory calculations, we propose that the oxygen activation pathway for the CO oxidation is initiated via an O$_2$-TiO$_2$ charge transfer complex directly excited by the 770 nm pump laser. Our results give insight into the fundemental understanding of photocatalytic processes of TiO$_2$ polymorphs relevant for the design of more efficient photoctalaysts.

Photocatalysts are promising materials to harvest solar energy[1] or purify polluted air and water[2]. Recent pandemics showcased the significance of the antibacterial and antiviral properties of photocatalytic surfaces[3,4]. One promising material is TiO$_2$, a widely applied photocatalyst with strong oxidizing properties. TiO$_2$ is low-cost, non-toxic[5], chemically and biologically stable, and shows antiviral and antibacterial properties[6–8]. The interest in TiO$_2$ as a photocatalyst increased in 1972 when Fujishima and Honda used a TiO$_2$ semiconductor photoanode for water splitting under UV light[9], and since then, this oxide was considered a component for solar cells[1]. Today, the photocatalytic properties of TiO$_2$ were studied for a range of reactions, such as water splitting[10] for hydrogen generation[11], and were tested in field studies for the degradation of pollutants in air and water[6,8], e.g., as an additive for concrete[12]. Rutile is the most stable polymorph of TiO$_2$ and is a widely studied model system for science on metal oxide surfaces[13]. Therefore, research on rutile contributes to a deeper understanding of the nature of photocatalysts.

One well-studied heterogeneous catalytic model reaction is CO oxidation due to its simplicity and its character as a benchmark system[14]. CO oxidation has a single product, CO$_2$. CO photooxidation on rutile and anatase TiO$_2$ was studied using Infrared Reflection Absorption Spectroscopy as well as X-ray Photoelectron Spectroscopy (XPS)[15–17]. As expected from studies on powdered samples[18], stoichiometric anatase (101) is the most photocatalytically active surface under UV-illumination, exhibiting faster reaction kinetics compared to reduced anatase (101) as well as reduced and stoichiometric rutile (110) in converting CO to CO$_2$. CO oxidation on both TiO$_2$ polymorphs was only observed in an O$_2$ atmosphere under concurrent UV-illumination.

In these experiments, the UV-illumination initiates a photocatalytic reaction, as an electron-hole pair is generated in the conduction and valence band[19]. Gas-phase O$_2$ dissociates by trapping the generated electron, resulting in adsorbed oxygen. The chemisorbed oxygen ion reacts with adsorbed CO to form CO$_2$[15]. The adsorption of oxygen is necessary for the reaction, as studies show, that lattice oxygen is not an oxygen source for this photoreaction[20]. The adsorption and photoactivation of oxygen is the crucial step, initiating the CO oxidation as an electron-mediated reaction, and thus competing with charge carrier recombination[21]. The efficiency of this process directly impacts the efficiency of the catalyst. A longer lifetime of charge carriers increases the probability of interacting with a gas-phase oxygen molecule. Different studies[16,22] found a shorter bulk lifetime of charge carriers in rutile compared to anatase. The reason is that anatase has an indirect

[1]Centre for X-ray and Nanoscience CXNS, Deutsches Elektronen-Synchrotron DESY, Hamburg, Germany. [2]The Hamburg Centre for Ultrafast Imaging, Hamburg, Germany. [3]Fachbereich Physik, Universität Hamburg, Hamburg, Germany. [4]Deutsches Elektronen-Synchrotron DESY, Hamburg, Germany. [5]Bremen Center for Computational Materials Science, Universität Bremen, Bremen, Germany. [6]Computational Science Research Center (CSRC), Beijing, China. [7]Computational Science Applied Research (CSAR) Institute Shenzhen, Shenzhen, China. [8]School of Science, Constructor University, Bremen, Germany. [9]Institute for Advanced Study, Changdu University, Chengdu, China. [10]Present address: Fraunhofer Institute for Solid State Physics IAF, Freiburg, Germany. [11]Present address: European XFEL, Schenefeld, Germany. ✉e-mail: heshmat.noei@desy.de; andreas.stierle@desy.de

band gap that inhibits electron-hole pair recombination and therefore enables a higher percentage of generated charge carriers to initiate this reaction pathway. The direct band gap of rutile results in a shorter lifetime of charge carriers, thus lowering the catalytic efficiency[16].

The studies on the lifetimes of charge carriers focus on the bulk properties of the materials leaving a knowledge gap of catalytic reactions that occur at the surface. The general observation of longer lifetimes of bulk electron-hole pairs in anatase is in alignment with the higher photocatalytic efficiency. Still, it ignores the influence of adsorbates on the catalytic surface under reaction conditions. In addition, the lifetimes of surface charge carriers may differ from bulk charge carriers and are influenced by band bending, defects, surface traps, and adsorbates[23]. It is therefore important to study the reaction dynamics in a catalytic environment to elucidate the reaction mechanism[21].

The ultrafast real-time dynamics during the CO oxidation on anatase $TiO_2$ was previously studied by Wagstaffe et al.[24] in a pump-probe experiment at FLASH. On anatase (101) CO photooxidation to $CO_2$ induced by a 770 nm laser was observed with a delayed onset between 1.2 and 2.8 (±0.2) ps after illumination. Based on Density Functional Theory (DFT) calculations an $O_2$-$TiO_2$ charge transfer (CT) complex was proposed, that enabled a direct charge transfer from the anatase-$TiO_2$ valence band to the $O_2$ molecular states in the bandgap. It was proposed, that the directly excited adsorbed oxygen dissociates and provides the oxygen adatoms for the CO oxidation. This indicated that charge transfer can occur on a faster timescale than previously reported[25,26].

To elucidate the role of the oxygen activation timescale for the difference in catalytic activity of $TiO_2$ rutile in comparison to anatase, we studied the dynamics of CO photooxidation on rutile (110) at the FEL FLASH at the Deutsches Elektronen-Synchrotron (DESY) in Hamburg. Free-electron Lasers with pulses in the femtosecond timescale allow the study of ultrafast surface dynamics and possible reaction intermediates that are observable on picosecond timescales[27,28]. Using superconducting RF accelerator technology, FLASH provides high-repetition rate photon pulses suitable to observe chemical dynamics with sub-picosecond temporal resolution[29,30]. The dynamics of the photoinduced CO oxidation to $CO_2$ was monitored on rutile (110) with a temporal resolution of 250 fs. The formation of $CO_2$ is observed within 200 to 800 fs after illumination with an optical laser with a wavelength of 770 nm. Time-Dependent Density Functional Tight-Binding (TD-DFTB) calculations showed the formation of an $O_2$-$TiO_2$ charge transfer complex as a possible pathway for ultrafast oxygen activation. Our results demonstrate that, although anatase is the more active photocatalyst compared to rutile, the dynamics of the CO oxidation on rutile is faster. The observation of different reaction dynamics on rutile and anatase is a further step to link the electronic structure of a material to its dynamics and the charge transfer to reactants.

## Results

In this experiment, we studied the ultrafast dynamics of the CO oxidation to $CO_2$ on rutile (110) in a controlled gas atmosphere of CO and $O_2$ each with a partial pressure of $3 \cdot 10^{-8}$ mbar at a sample temperature of 80 K. An optical pump laser (770 nm/1.6 eV) and the third harmonic of the FEL (hν = 643 eV) as a probe beam were spatially and temporally overlapped. The relative timing between the pump and probe pulses was controlled by a mechanical delay stage as part of the optical laser setup. The rutile $TiO_2$(110) sample surface was prepared as described in the experimental section. With the FEL the O 1s, Ti 2p, and C 1s core levels were probed. Because of the increasing amount of water on the surface during the experiment, data were analyzed by integration over the first 15 min after flash-annealing the sample to observe the intrinsic behavior of the system, as displayed in Fig. 1.

As a prerequisite for the pump-probe experiments, the temporal resolution could be determined to be 250 fs. For this, the Full Width Half Maximum (FWHM) of the sidebands at zero delay in the Ti 2p (Fig. S1) and O 1s (Fig. S2) core level spectra have been evaluated. Sidebands appear when the optical laser and the FEL are temporally and spatially overlapped, and are represented as a replica of the original photoemission line shifted by the

energy of the optical laser through absorption or stimulated emission[31,32]. The Ti 2p spectra (Fig. S1c) exhibit a small shoulder on the lower binding energy side of the $Ti^{4+}$ $2p_{3/2}$ lattice peak, assigned to $Ti^{3+}$ surface defects. The area of the $Ti^{3+}$ $2p_{3/2}$ peak amounts to 5 ± 1% compared to the $Ti^{4+}$ contribution. As one single oxygen vacancy contributes to two $Ti^{3+}$, the estimated amount of oxygen vacancies at the surface is 2.5 ± 0.5%[33]. Defects can also occur in the form of $Ti^{3+}$ interstitials.

In prior XPS experiments, the kinetics of CO oxidation to $CO_2$ on rutile $TiO_2$ (110) under UV light (365 nm) was studied with an X-ray laboratory source (hν = 1486.6 eV) at the DESY Nanolab[34]. In the O 1s core level region, adsorbed CO gives rise to a component at a binding energy of 536.1 eV and $CO_2$ at 535.0 eV on rutile (110) with a difference of 1.1 eV between the two components (Fig. S3). The assignment of the binding energies of the O 1s core level of CO and $CO_2$ in this study is in agreement with previous results[17]. The CO peak arises at 536.5 eV on rutile (110) at 80 K in $3 \cdot 10^{-8}$ mbar CO atmosphere. At this temperature, it was reported that half a monolayer of CO adsorbs on rutile (110)[35]. Slight differences in the binding energy are expected due to coverage-dependent shifts[36]. Here we monitored the ultrafast dynamics of the conversion of CO to $CO_2$ in the oxygen O 1s core level. The O 1s core level was chosen to monitor the time-resolved CO oxidation, as the favorable stoichiometry of the reaction product $CO_2$, consisting of two oxygen atoms, results in higher absolute count rates. Additionally, the cross-section for photoionization at 643 eV is higher for the O 1s region than the C 1s region[37]. To obtain enough statistics to extract the time-resolved data shown in Fig. 1, the data were binned by steps of 200 fs. The detected photoelectrons of the O 1s and Ti 2p core level region have a kinetic energy of 100 eV and 175 eV, respectively, and the inelastic mean free path (λ) of $TiO_2$ for electrons at those energies is calculated to be 5.6 Å and 6.8 Å[38]. The XPS probing depth, from which 95% of the measured photoelectron originates, corresponds to 3λ[39]. In $TiO_2$, which has a layer spacing of 3.25 Å in the [110]-direction[40], the photoelectrons measured for

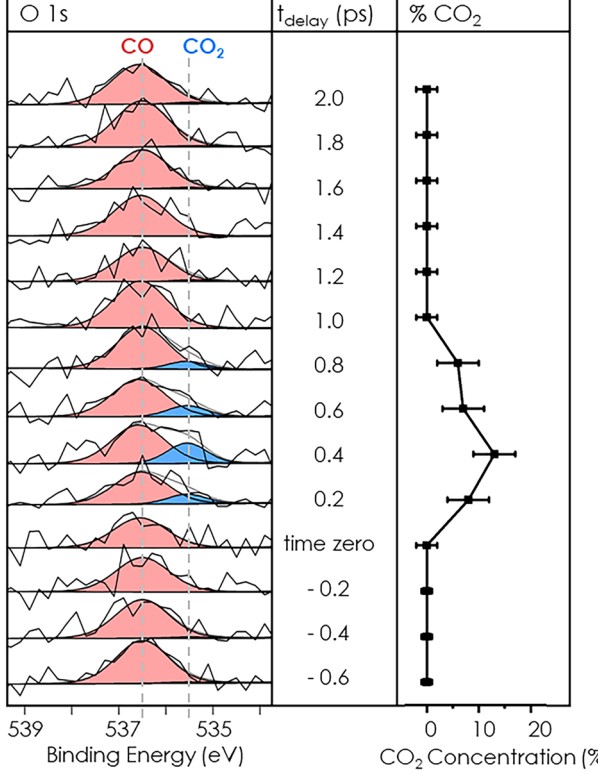

**Fig. 1** | Left panel: Deconvoluted time-resolved O 1s core level spectra binned by 200 fs steps reveals ultrafast CO (red) oxidation to $CO_2$ (blue) on rutile (110) at 80 K at FLASH. Right panel: $CO_2$ concentration obtained from the integrated O 1s core level. The data were taken within the first 15 min after flash-annealing the surface.

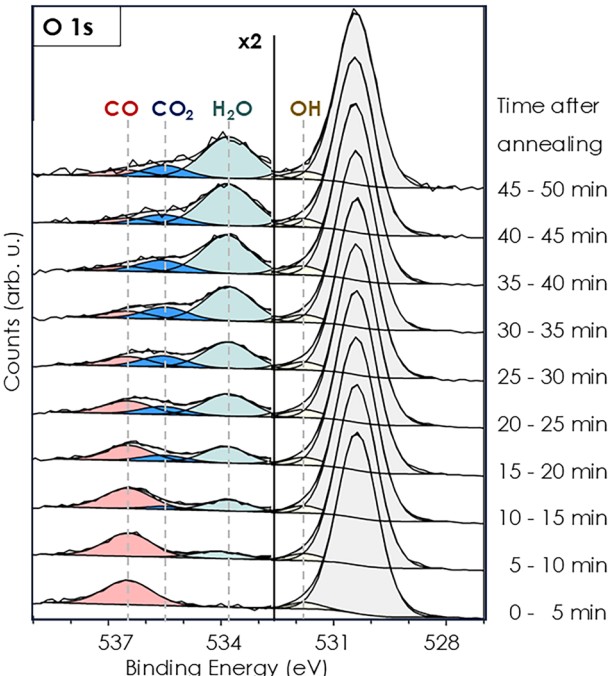

**Fig. 2 | Deconvoluted O 1s core level average spectra during CO oxidation at FLASH binned in 5 min steps after heating and cooling the sample to 80 K.** Adsorbed CO (red), produced $CO_2$ (blue) and adsorbed water (light blue).

the Ti 2p region originate from the first 6.3 atomic layers and for O 1s from the first 5.2 atomic layers. This emphasizes the high surface sensitivity of this technique.

XP spectra of the O 1s core level were recorded within a delay range from −2 ps before to 12 ps after optical excitation. The results presented in Fig. 1 and discussed in detail in this work were only recorded within the pump-probe delay range from −0.6 to 2.0 ps in which the $CO_2$ formation was observed. After the sample preparation, data were collected for 50 min. In the range of the first 800 fs after the pump laser initiates the reaction, a peak shoulder located at 535.5 eV appears, which is assigned to the formation of $CO_2$[17]. The maximum amount of $CO_2$ is detected at 400 fs with 13 ± 4% normalized to the CO signal. The complete binding energy and delay range from −2 to 12 ps is shown in Fig. S4 as spectra and in Fig. S5 as a XP color map. No other $CO_2$ signal was resolved in that delay range up to 12 ps within the sensitivity of our experiment.

The data averaged over the whole delay window in 5 min bins after the flash-annealing is shown in Fig. 2. During data acquisition at 80 K in a CO/$O_2$ atmosphere with a partial pressure of $3 \cdot 10^{-8}$ mbar for CO and $O_2$ each, residual water in the UHV system of the experimental chamber accumulates on the surface (residual pressure: $4 \cdot 10^{-10}$ mbar). The peak at 534 eV is assigned to water[41] and it increases with time. Water partially dissociates to hydroxyls (OH) on the rutile surface and appears as a shoulder of the lattice $O^{2-}$ peak at 531.8 eV[42]. This shoulder is already visible within the first 5 min and before the $H_2O$ peak at 534 eV appears. The OH signal increases with the water coverage. Water impedes the adsorption of CO and promotes the stabilization of adsorbed $CO_2$[43]. As $H_2O$ binds more strongly to the rutile (110) surface than CO, the CO peak decreases over time as $H_2O$ blocks adsorption sites and inhibits readsorption of CO from the gas-phase as seen in the time-averaged data (Fig. 2). The data collected in the first 5 min in the absence of adsorbed water does not offer sufficient statistics for time-resolved binning. In the first 15 min, used for the time-resolved analysis in Fig. 1, a low amount of water and a negligible amount of $CO_2$ is adsorbed on the surface. In the corresponding time-resolved data in Fig. S4 the water signal is not visible because the high statistical noise of the small signal. This is in contrast to the spectra from 15 to 50 min, which show a clear $CO_2$ signal as a broad shoulder and a further increasing water signal, indicating that the

surface is covered with a non-negligible amount of $CO_2$ and $H_2O$. After 40 min the $CO_2$ signal does not increase anymore, indicating that no further CO is oxidized. Water accumulates further on the surface, blocking the adsorption sites for CO and $O_2$. The $CO_2$ formation was observed using the time-resolved spectra within the first 800 fs with a binning of 200 fs recorded in the first 0–15 min after flash-annealing. For comparison, the data for 0–15 min and 15–30 min after heating were binned in 500 fs time windows and $CO_2$ formation at 535.5 eV was observed for both data sets in the spectra 250–750 fs after time zero as seen in Fig. S6.

In the O 1s spectra, no peak can be assigned to adsorbed atomic or molecular oxygen under the experimental conditions of $3 \cdot 10^{-8}$ mbar $O_2$ and $3 \cdot 10^{-8}$ mbar CO at 80 K. Physisorbed $O_2$ was observed below 60 K on anatase (101) with a binding energy of 537.3 eV as a double peak in a triplet state[44]. On stoichiometric rutile $TiO_2$(110), molecular oxygen only interacts weakly with the surface and physisorbs at low temperatures below 85 K[45]. When oxygen vacancies are present, $O_2$ chemisorbs on the surface in a peroxo $O_2^{2-}$ state in the oxygen vacancy itself or in the direct vicinity, on top of a five-fold coordinated Ti ($Ti_{5c}$) atom. Even at low temperatures below 80 K, adsorbed $O_2$ can heal an oxygen vacancy leaving oxygen adatoms on the $TiO_2$ surface[46]. Due to defects (oxygen vacancies) of 2.5%, chemisorption of $O_2$ is probable on the surface but due to the very low amount, it is below the detection limit in the O 1s spectra. Note, that the $O_2$ signal overlaps with the strong O 1s signal from CO at 536.5 eV, making it impossible to detect it independently in our time-resolved measurements.

To explain the experimental observation of ultrafast CO oxidation within 800 fs, we performed first principle TD-DFTB calculations for the adsorption of $O_2$ and CO on the rutile $TiO_2$(110) surface. The optimized geometry of CO and $O_2$ adsorption can be found in Fig. S7 and the density of states (DOS) of these systems in S8. Two favorable configurations are found for the coadsorption of CO and $O_2$. In both configurations, the CO molecule interacts via the carbon atom with $Ti_{5c}$ in agreement with previous reports[45,47,48]. The $O_2$ molecules either adsorb perpendicular or parallel to the surface on top of the neighboring $Ti_{5c}$ site with a calculated adsorption energy of −0.667 eV for the perpendicular and −0.505 eV for the parallel $O_2$ + CO configuration, respectively. The DOS calculations reveal that the presence of adsorbed $O_2$ molecules is related to the appearance of electronic states in the band gap of the surface model, which are responsible for a new band at lower energies in the absorption spectrum. This indicates the formation of an $O_2$-$TiO_2$ CT complex that is activated by visible/infrared light via a direct electron transfer from the $TiO_2$ valence band to adsorbed $O_2$ molecules. Such a charge transfer complex was proposed previously by Wagstaffe et al.[24] for the ultrafast CO oxidation on anatase initiated by a 770 nm laser and by Freitag et al.[49] for the visible light activity of $TiO_2$ with adsorbed nitrogen(II) oxide. In this study, we additionally calculated the absorption spectrum of $TiO_2$ with and without adsorbed $O_2$ using real-time TD-DFTB implementation. The absorption spectra are shown in Fig. 3. Stoichiometric $TiO_2$ has no adsorption bands in the visible light region, but upon $O_2$ adsorption new adsorption bands in the visible light region appear due to the formation of a CT complex. Via the CT excitation, adsorbed $O_2$ is reduced to $O_2^-$ which is the initial step of $O_2$ dissociation.

## Discussion

In this experiment, the oxidation of CO to $CO_2$ is observed within the first 800 fs after initiation. Figure 4 illustrates the reaction mechanism. After 800 fs $CO_2$ desorbs and new $CO_2$ forms only in the next cycle (see Fig. 1). This gives evidence, that $O_2$ dissociates on $TiO_2$ by a light activated process as $O_2$ only physisorbs on stoichiometric $TiO_2$ at 80 K and does not dissociate. On the other hand, on defective rutile (110) oxygen adatoms were observed in small concentrations at 80 K after $O_2$ adsorption and healing of an oxygen vacancy which leaves an isolated oxygen adatom[46]. However, this mechanism can be excluded for a reaction cycle, as the oxygen vacancy is healed after the $O_2$ dissociative adsorption. Other $CO_2$ signals were not detected in the 12 ps delay window, which would indicate another CO oxidation pathway with a different timescale (see Fig. S4). The $CO_2$ signal,

therefore, implies that oxygen was reduced, dissociated, and reacted with CO to $CO_2$ within 800 fs.

For CO photooxidation on $TiO_2$, the activation of oxygen is considered the rate-defining step. This includes charge transfer to oxygen and dissociation of the anion. To generate electron-hole pairs on $TiO_2$, photons with either an energy larger than the bandgap or, in case of photons with a smaller energy, multi-photon absorption is required to excite electrons from the valence to the conduction band. Multi-photon absorption to match the bandgap of 3 eV for bulk rutile $TiO_2$[50] was reported for laser pulses with pulse energies up to 3 μJ and wavelengths of 774, 800, and 813 nm from a chirped-pulse amplified Ti:Sapphire system[51]. The optical laser fluence used in this study fits within those parameters, having a pulse energy of 5–10 μJ and a wavelength of 770 nm. The excited electrons in the conduction band can either recombine, be trapped, or induce an oxidation/reduction pathway, in this case, the reduction of $O_2$. Photogenerated electrons in the $TiO_2$ band structure are generated in less than 100 fs and conduction band electrons are trapped at the surface within 200 fs[52]. The transfer of surface-trapped electrons to oxygen was only observed on a nanosecond timescale in less than 100 ns[25] and is significantly faster than the transfer of conduction band electrons to oxygen in 10–100 μs[53]. In this activation pathway, the electron-driven oxygen activation competes not only with charge carrier recombination but also with the hole-driven desorption of oxygen[54].

Alternatively, based on our DFT calculations, an $O_2$-$TiO_2$ CT complex is proposed. Adsorbed oxygen on stoichiometric rutile (110) introduces unoccupied states into its band gap with excitation energies relating to the visible light range as seen in Fig. 3. The direct excitation of the oxygen molecule may activate oxygen faster than by photogenerated conduction band electrons. As a result, the activated $O_2^-$ dissociates and reacts with CO to $CO_2$. The activation with a 770 nm laser only requires one photon absorption to excite the $O_2$-$TiO_2$ CT complex. The CT complex's excitation probability is lower than band-to-band excitations since the excitation is limited by the number of acceptor states offered by the adsorbed oxygen molecules. But the CT complex excitation is more efficient since the charge is directly trapped in the reduced oxygen molecule and does not compete with recombination. A similar charge transfer complex was proposed for NO on $TiO_2$ for the visible light for the degradation of NO[49]. The photonic efficiency under UV (bandgap excitation) and visible light (CT complex excitation) are in the same order of magnitude. However, the CT process is one order of magnitude less likely to occur. The CT pathway does not compete with charge carrier recombination and thus increases the photonic efficiency of the desired reaction.

The $O_2$-$TiO_2$ CT is also a possible mechanism for anatase $TiO_2$(101). In a similar study at FLASH in which the ultrafast CO oxidation on anatase (101) at 60 K was investigated, the $CO_2$ formation was assigned to the direct activation of oxygen[24] and was observed from 1.2 ps to 2.8 ps after initiation. In contrast, this study on rutile (110) at 80 K reveals that $CO_2$ is observed already at 0.2 ps after initiation. The temperature difference might influence the diffusion of the adsorbates on the surface, thus enabling a faster reaction between CO and $O_2$[55]. In our experiment, measurements at 60 K were not feasible due to rapid water adsorption on rutile (110), which blocked adsorption sites for the reactants. Therefore, we conducted measurements at 80 K instead.

In comparison, CO oxidation was observed in the first few picoseconds after initiation on catalytic metals such as Ru(0001)[56,57], Pt(111)[58] and Pd(111)[59]. Important to note is, that only atomic and no molecular oxygen was adsorbed on these metal surfaces. Furthermore, the oxygen was not dissociated during the reaction process. The role of the optical laser was to excite electrons in the substrate, resulting in an energy transfer to the adsorbate. This causes vibrational motions that induce the reaction between CO and O. Öström et al.[57] studied the ultrafast CO oxidation on Ru(0001) with femtosecond X-ray laser pulses and reported the activation of O within 300 fs, of CO in 500 fs, and a formation time for $CO_2$ of 800 fs. In our experiment we observe a faster $CO_2$ formation onset time hinting at a different reaction mechanism. As mentioned above, atomic oxygen on rutile-$TiO_2$(110) is only observed on reduced rutile at surface oxygen vacancies. Molecular oxygen chemisorbs on top of an oxygen vacancy, heals the vacancy by dissociation, and leaves an O adatom[46].

We argue that atomic oxygen on oxygen vacancies is unlikely to be responsible for the fast CO oxidation. The low amount of 2.5 % oxygen vacancies in the Ti 2p spectra is in disagreement with the maximum of $13 \pm 4\%$ $CO_2$. The amount of oxygen vacancies indicated by $Ti^{3+}$ does not change during the CO oxidation (see Fig. S9). It also indicates that another CO oxidation mechanism without defect contribution is present, since the CO oxidation is also observed on the stoichiometric rutile (110) and anatase (101)-$TiO_2$ surfaces[17]. Oxygen adatoms are observed to be unstable on anatase (101), so the mechanism including complexes of CO and oxygen adatoms is not a possible pathway on anatase[60].

The rutile (110) and the anatase (101) surface results differ in three aspects: in the amount of observed $CO_2$ compared to CO, the time between initiation and first $CO_2$ signal detection, and the decay time of the $CO_2$ peak.

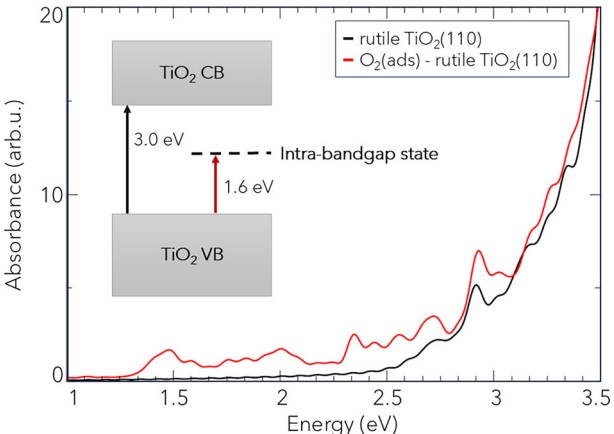

**Fig. 3 | DFT calculation of an absorption spectrum of $TiO_2$ before (black) and after (red) $O_2$ adsorption.**

**Fig. 4 | Schematics of the time-resolved reaction mechanism, see text for details.**

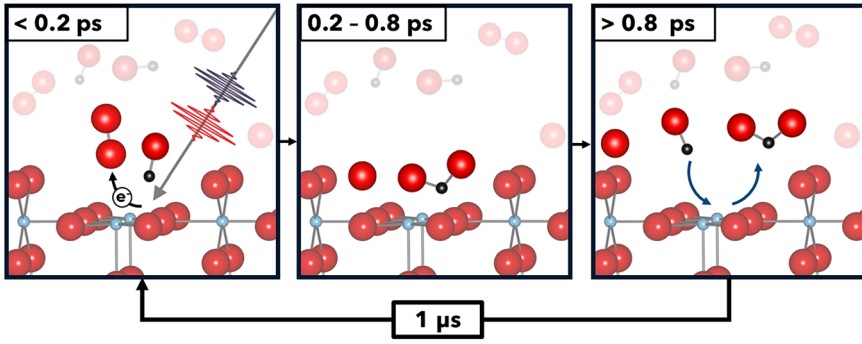

The $CO_2$ signal on anatase (101) was detected for 1.6 ps with a maximal $CO_2$ concentration of ~25 % relative to the CO signal. After 2.8 ps the formed $CO_2$ was desorbed. For comparison, on rutile (110) no $CO_2$ is observed after 1 ps and the maximum relative $CO_2$ concentration was ~13% after 0.4 ps. The higher relative $CO_2$ concentration aligns with the observation that anatase (101) shows higher photocatalytic activity than rutile (110) and it implies that the charge transfer process is more efficient on the anatase (101) surface.

One major argument found in the literature for the higher photo-catalytic activity of anatase is the charge carrier lifetime difference in rutile and anatase, which were studied for single-crystals and powders[16,22,23,61,62]. However, those measurements always detected the concentration of bulk charge carriers and not $e^-/h^+$ at surface sites for charge transfer to adsorbates. In stoichiometric rutile, charge carrier recombination is more likely and charge carriers are shorter-lived than in stoichiometric anatase. Maity et al.[22] found using transient absorption spectroscopy, that bulk charge carriers decaying in ~0.5 ps for stoichiometric and reduced rutile (110) single crystals, whereas in stoichiometric and reduced anatase the lifetime was 32 ps and 24 ps, respectively. The timescale for rutile in that study is similar to the timescale of the $CO_2$ formation in our study. The longer lifetime of charge carriers in anatase than rutile is in agreement with measurements from Xu et al.[16] who observed a direct band gap with faster charge carrier recombination for rutile and an indirect bandgap for anatase with inhibited charge carrier recombination which links the bulk properties of rutile and anatase to its photocatalytic performance. In an indirect bandgap semiconductor, recombination requires a phonon to conserve momentum due to the mismatch between the valence band maximum and conduction band minimum[63] and results in a prolonged lifetime of charge carriers. But the lifetime of surface charge carriers depends highly on surface adsorbates and not only on bulk properties[19]. Adsorbates can not only act as traps for photogenerated charge carriers but also induce band bending, promoting the migration of charge carriers to the bulk. The separation of charges directly influences the charge carrier lifetimes[23]. In this experiment, the surface dynamics are measured on a picosecond timescale, in contrast to the bulk lifetimes of charge carriers of several nanoseconds. The decay of the $CO_2$ signal within 1 ps can also indicate that the direct excitation of $O_2$ via the CT complex and not charge carriers from the bulk are responsible for the ultrafast oxidation of CO observed in this study. The reaction time is therefore dependent on the lifetime of the excited oxygen. We cannot exclude a further oxygen activation mechanism mediated by trapped charge carriers from the $TiO_2$ conduction band on a nanosecond timescale, which is not detected in our study.

It is important to note that, although the surface is not deliberately exposed to water in this experiment, a growing water peak is observed after 5 min at 80 K in the $CO/O_2$ atmosphere. Since the time-resolved data requires 15 min of each run for sufficient statistics water is present in the time-resolved spectra but in too low amount for a quantitative analysis. On rutile (110) water adsorbs on top of $Ti_{5c}$[64] and partially dissociates into hydroxyl (OH) groups. Water influences CO oxidation and can promote or inhibit the reaction. In one study[65], the CO oxidation rate under UV light increased until a coverage of up to 1/2 ML of water was reached and decreased for higher coverages. It was proposed that on rutile (110) under UV light in the presence of water $H_2O_2$ as well as surface peroxo-species such as Ti–O–O–H and Ti–O–O–Ti are formed[66]. The CO oxidation in the presence of water appears to correlate with the amount of peroxide species formed. Several studies[65–69] agree, that water blocks the adsorption sites for CO decreasing CO adsorption and therefore decreasing the $CO_2$ formation rate. We also observe a decrease in CO adsorption with increasing water adsorption (see Fig. 2). Water could also influence the activation of oxygen and the charge transfer from $TiO_2$ to $O_2$ as the initial step of the CO oxidation. Wagstaffe et al.[41] reported the ultrafast hole transfer from anatase-$TiO_2$(101) to water within 285 fs as well as the formation of a hydrogen bond between water and the $O_{2c}$ site.

Tilocca et al.[70] investigated the adsorption of $O_2$ on the hydroxylated rutile (110) surface with molecular dynamics simulations. Physisorbed $O_2$ can interact with OH-groups without going through a chemisorbed state. The adsorption structures included hydrogen bonds between chemisorbed $O_2$ and OH, structures resulting from proton transfer as the formation of hydroperoxyls ($HO_2$), and hydrogen peroxide ($H_2O_2$), and less stable structures resulting from dissociative $O_2$ adsorption as $OH_t$ and $O_a$. Hydroperoxyls $HO_2$ and bridging hydroxyls $OH_t$ were observed experimentally by Scanning Tunneling Microscopy[71,72], Kelvin Probe Force Microscopy, and Atomic Force Microscopy[73]. Molecularly chemisorbed $O_2$ next to $OH_{br}$ was not observed. Calculations found an increased $O_2$ adsorption mediated by adsorbed OH groups due to a charge transfer from OH to $TiO_2$[74]. Upon adsorption, the charge is transferred to $O_2$ thus stabilizing the adsorption. Local Contact Potential Difference measurements suggest experimental evidence for the charge transfer from $Ti_{5c}$ atoms to oxygen $O_a$[73]. More recent calculations confirmed that $O_2$ adsorption is favored on rutile (110) in the presence of OH groups[75] and that the energy barrier for the O=O scission, necessary for the CO oxidation, is lowered by proton transfer, which is induced by adsorbed water[76]. The interaction of $OH/H_2O$ with $O_2$ might thus facilitate the interfacial charge transfer, leading to a enhanced $O_2$ and CO interaction, and therefore $CO_2$ formation. We cannot exclude, that the faster $CO_2$ formation on rutile $TiO_2$(110) compared to anatase $TiO_2$(101) could be due to the presence of water.

## Conclusion

In conclusion, we investigated the dynamics of the CO oxidation on rutile $TiO_2$(110) by optical pump, FEL probe X-ray photoemission spectroscopy. In an $O_2/CO$ atmosphere at 80 K, CO adsorbs on the rutile (110) surface and is oxidized to $CO_2$ within the first 800 (±200) fs after excitation by the 770 nm laser. We propose, that $O_2$ adsorbs molecularly on the surface and is activated via an $O_2$-$TiO_2$-CT complex. Residual water in UHV blocks CO adsorption sites and reduces the $CO_2$ oxidation but might in low coverages facilitate charge transfer. With time-resolved XPS, several oxygen-containing components in the O 1s core level were monitored simultaneously, allowing studying reaction dynamics of co-adsorbed reactants or several products non-destructively in real-time.

While on anatase $TiO_2$(101) the CO oxidation is observed within 1.2 and 2.8 ps after initiation, the $CO_2$ signal on rutile is visible between in the first 0.8 ps. This indicates a shorter activation time of the oxygen species on rutile (110) likely related to a faster charge transfer. Although anatase is the more active photocatalyst compared to rutile, the dynamics of the CO oxidation on rutile is observed to be faster. The observation of different reaction dynamics on rutile and anatase is a further step to link the electronic structure of a material to its dynamics and the charge transfer to reactants. Our study sets the experimental basis for future time resolved full band structure theoretical studies which will further elucidate the mechanistic origin of the different reaction dynamics. Such a deeper understanding will help in tailoring photocatalytic systems which is crucial for developing more efficient materials for green energy production, as water splitting or photovoltaics.

## Methods
### Experimental
The time-resolved photoemission data were taken at the plane grating monochromator beamline PG2[77,78] of the free-electron laser FLASH[29,30] located at DESY in Hamburg, Germany. The fundamental wavelength of FLASH was 5.79 nm (214 eV) with a pulse energy of 25–40 µJ. To probe the core level of oxygen O 1s the monochromator was tuned to the third harmonic of 1.93 nm (643 eV). The FEL pulses were delivered with a macro-bunch repetition rate of 10 Hz with each macrobunch consisting of 400 bunches with a 1 MHz repetition rate. The temporal FWHM of each FEL pulse was <100 fs, though stretched in the monochromator to 150 fs. The optical pump laser with a wavelength of 770 nm (1.6 eV) matches the pulse pattern of the FEL. The maximum single pulse energy of the optical laser was 5-10 µJ with a spot size of ~300 µm under normal incidence. The fluence could be calculated to be $7-14 \, mJ \cdot cm^{-2}$ under the measurement geometry of 55° sample tilt with respect to the incoming laser beam. The laser beam is

coupled in collinearly with the FEL. To prevent CO desorption the laser fluence was attenuated to 2.2 mJ cm$^{-2}$. The temporal FWHM of the optical laser pulse was ~120 fs. A mechanical delay stage set the temporal delay of the optical laser with respect to the FEL beam. The experimental setup used at the beamline was the wide-angle electron spectrometer (WESPE)[79] chamber. WESPE consists of a sample preparation chamber with an ion gun, heating station, and low energy electron diffraction (LEED). The main experimental chamber is equipped with a Themis 1000 high-resolution time of flight spectrometer with a three-dimensional delay line detector (3D-DLD4040-4Q, Surface Concept), beamline connection, and leak valves for dosing gases. The spectra were recorded with a pass energy of 20 eV. The used gases were Ar (purity 99.999%) for sample preparation and CO (purity 99.97%) and O$_2$ (purity 99.999%). The cryostat in the manipulator, which holds the sample under investigation, allowed cooling with liquid He. The rutile TiO$_2$(110) single crystal (7 mm × 7 mm × 1 mm) was cleaned under ultra-high vacuum (UHV) conditions with a base pressure of $3 \cdot 10^{-10}$ mbar by repeated cycles of 1 keV Ar$^+$ ion sputtering and flash-annealing to 650 °C (for less than 1 min) and cooled in $1 \cdot 10^{-6}$ mbar O$_2$ until a (1 × 1) LEED pattern was obtained (Fig. S11). X-ray photoelectron spectra confirmed the absence of carbon contaminations (Fig. S10). Although the sample was annealed and cooled in oxygen, the Ti 2p core level spectra show 4–6% of Ti$^{3+}$ as a small shoulder next to the lattice peak of Ti$^{4+}$, as seen in Fig. S1c. Ti$^{3+}$ indicates defects in the form of oxygen vacancies[80]. During the CO oxidation, the sample was cooled by liquid He to 80 K and was held in an atmosphere of CO and O$_2$ with partial pressures of both $3 \cdot 10^{-8}$ mbar. To avoid potential laser-induced damage, the incident pulses were scanned across the sample surface. During data acquisition at 80 K in a gas atmosphere of CO and O$_2$ each with a partial pressure of $3 \cdot 10^{-8}$ mbar, residual water from the UHV environment adsorbed on the cold sample surface resulting in a growing peak at 534.8 eV, as seen in Fig. 2. To limit the influence of water on the reaction dynamics, only spectra recorded until 15 min after brief flash-annealing of the surface to 600 K are used to study the dynamics of the CO oxidation. The sample was cleaned by sputtering and annealing in O$_2$ after two measurement cycles with flash-annealing to obtain a stoichiometric surface. The binning of the extracted spectra was were 200 fs in temporal domain and 200 meV in electron energy. In total, the data of the first 10–15 min of 22 runs was used, depending on the amount of adsorbed water. Due to shifts of the FEL photon energy, the Ti 2p spectra of each run were calibrated by aligning the Ti$^{4+}$ 2p$_{3/2}$ to 459.0 eV and the O 1s spectra by calibration the lattice O$^{2-}$ to 530.4 eV[13]. For each run time zero, the temporal overlap of FEL and optical laser was determined by fitting the sidebands of the O 1s lattice peak. The spectra were fitted in CasaXPS with Gaussian/Lorentzian curves on a Shirley or linear background. The Shirley background emulates the inelastic electron scattering of the intensive O 1s lattice peak. Regions with lower counts were fitted with a linear background as the modeling of the inelastic scattering did not improve the fit (see Supplementary Data).

### Theroretical

The DFT periodic calculations on the neutral TiO$_2$ rutile (110) surface were performed with the Vienna ab initio simulation package (VASP) code[81–84]. The TiO$_2$ rutile (110) surface was modeled by a slab model of 40 Ti atoms and 80 O atoms using the following lattice parameters: $a = 5.9612$ Å, $b = 13.0834$ Å, and $c = 30.0000$ Å. For the optimization of the structures with the Perdew-Burke-Ernzerhof (PBE) functional[85], the projecto augmented-wave method[86,87] was used and the Brillouin zone was sampled with a (2 × 2 × 1) Monkhorst-Pack k-points grid with an energy cutoff of 400 eV. In the systems with O$_2$ molecule presence, spin polarization calculations are performed to include the triplet nature of the O$_2$ molecule in the ground state. Van der Waals interactions were included by using the DFT-D3 dispersion corrections with Becke-Johnson damping[88,89]. All minima were confirmed by frequency calculations. The adsorption energies $E_{ads}$ were calculated by: $E_{ads} = 1/n \left( E_{complex} - E_{TiO_2} - nE_{mol} \right)$ where: $E_{complex}, E_{TiO_2}, E_{mol}, n$ are the total energy of the molecule-TiO$_2$ complex formed by molecule-TiO$_2$ rutile (110) surface, the TiO$_2$ rutile (110) surface, the

molecule and number of molecules, respectively. Within this definition of adsorption energy, a negative value indicates an exothermic process. For the DOS analysis a (4 × 4 × 1) k-points grid, PBE functional[85] at DFT level of theory with the semi-empirical nonlocal external potentials was used[90–92]. The absorption spectra of O$_2$ were calculated using DFTB+ code[93]. The repulsive potential for the Ti-O pair was improved for a better description of the physisorption of O$_2$ over TiO$_2$ rutile (110) surface. The set of DFTB parameters *tiorg-0-1*[94] were modified and used. To determine the absorption spectra at the real-time TD-DFTB level, a cluster model was used to simulate the TiO$_2$ rutile (110) surface. The cluster is formed by Ti$_{21}$O$_{68}$H$_{52}$ formula, in which the peripheral O atoms were saturated with H atoms to keep the cluster neutral. To obtain the absorption spectra, an initial perturbation to the initial ground-state matrix is introduced. This perturbation has the shape of a Dirac delta pulse, and the density matrix evolves in time. Its evolution can be resolved by time integration of the Liouville-von Newmann equation of motion. For this, an initial electric field of 0.001 V/Å was used.

### Data availability

The XPS data of presented results in the paper and the SI are available as .vms file in the Supplementary Data.

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

## Acknowledgements

This work is supported by the Cluster of Excellence 'CUI: Advanced Imaging of Matter' of the Deutsche Forschungsgemeinschaft (DFG)—EXC 2056—project ID 390715994. We acknowledge DESY (Hamburg, Germany), a member of the Helmholtz Association HGF, for the provision of experimental facilities. Beam time at FLASH was allocated for proposal F-20190739.

## Author contributions

H.G., M.W., L.W., S.C., S.P., S.D., D.K., M.H., G.B., H.R., F.P. and H.N. participated in FLASH beam time and performed experiments. A.D.-C., V.G., A.D.G. and T.F. carried out DFT periodic calculations. H.G., L.W. and H.N. performed data analysis. H.G. and H.N. wrote the initial manuscript. All authors discussed the results and contributed to editing the paper. T.F. supervised DFT periodic calculations. H.N. and A.S. supervised the project. A.S. conceptualized the study together with H.N. and raised funding.

## Funding

## Competing interests

The authors declare no competing interests.
