## [Transparent Peer Review file · Communications Chemistry]

Dynamics of CO Photooxidation to CO₂ on Rutile (110)

Corresponding Author: Dr Heshmat Noei

Editorial note: Reviewer #2 co-reviewed the manuscript with one of the other reviewers, who provided a joint report.

Version 0:

Reviewer comments:

Reviewer #1

(Remarks to the Author)

This manuscript by Gleißner et. al. titled Dynamics of CO photooxidation to CO₂ on Rutile (110) is a study on ultrafast kinetics of CO oxidation monitored by XPS on a FEL. The scope of the manuscript is similar to a previous study presented by the same group on dynamics of CO photooxidation on anatase (101) surfaces using same experimental methodologies. The manuscript is well presented, the conclusions are supported by appropriate experimental observations and the discussions are at appropriate level.

I can recommend the manuscript for publication.

I have some minor comments and question, mostly to satisfy my own curiosity.

1) Can the authors comment on why the study was conducted at 80 K instead of at higher temperatures (room temperature for example)? It seems that water adsorption onto the rutile surface (due to cryogenic trapping) is a significant problem. For instance, this study was conducted at 80 K, due to significant water adsorption at 60 K, while analogous experiment reported for anatase was at 60 K. This makes direct comparison of the two data sets more complicated. Wouldn't studies at higher temperature negate this problem?

2) In addition, the authors also discussed the potential role of water towards the reaction mechanism. Given the non-observable H₂O signal at the ps regime, what are the implications towards the reaction mechanism?

3) Can the authors comment on the lack of observable signal that can be attributed to physisorbed, chemisorbed molecular O₂ or reduced O₂ on rutile surface (and anatase surface from previous studies)? One of the conclusion of the manuscript is that the fast kinetic observed can be rationalized by a O₂-TiO₂ charge transfer complex. Why this O₂-TiO₂ CT complex is not observable by XPS?

4) For spatial and temporal alignment of the optical laser and FEL pulse, the authors monitor the formation of side bands at 459 eV. In Figure S3, it seems that these side bands occur at t = -0.2 ps delay. Is this because of the relatively large jitter time of the optical laser pulse or an artifact of binning?

5) Data was collected by raster scanning across the rutile surface to minimize radiation damage (by FEL pulse and also desorption of CO by optical laser pulse). The repetition rate of the FEL pulse is very high at 1 MHz. It seems to me that the sample cannot be feasibly moved to a fresh spot for each shot (speed in the order of 100 m/s). Can the authors comment on how significant is the radiation damage with this current setup and implication to data quality and analysis?

Reviewer #2

(Remarks to the Author)

Reviewer #3

(Remarks to the Author)

The dynamics of CO oxidation on rutile TiO₂(110) was investigated using free-electron laser X-ray

photoemission spectroscopy at 80 K. Building upon a previous study employing the same technique, which observed CO₂ formation on anatase TiO₂(101) between 1.2 and 2.8 ps after time zero, this work demonstrates a similar phenomenon on rutile but on a faster timescale (0 to 0.8 ps). I appreciate the work done by the authors since the experimental results combined with the theory highlights a different reaction mechanism. However, I have few major concerns which I suggest being resolved for making the manuscript more cogent, understandable to a wide range of readers and shed light onto the importance of this work. Please see the pdf for detailed comments.

Reviewer #4

(Remarks to the Author)

Report on the manuscript entitled "Dynamics of CO Photooxidation to CO₂ on Rutile (110)" by Helena Gleißner et al.

*** General comments ***

The manuscript presents a very detailed study on the ultrafast photoinduced oxidation dynamics of CO to CO₂ on the surface of rutile TiO₂ catalysts. This work is essentially a continuation of the authors' previous research on the same mechanism in anatase TiO₂, which was published in the literature a few years ago.

This type of investigation, which combines both experimental and theoretical innovative methods, is highly timely and has a significant impact on various fields, including research on sustainable energy sources.

I enjoyed reading the manuscript; however, at times, I had the impression that some sections are somewhat redundant, and the level of detail is excessive. I encourage the authors to revise the text, moving non-essential parts into the Methods and Theoretical sections. This would enhance readability and improve the overall flow of the manuscript.

More importantly, the authors should better articulate their motivation for investigating the rutile TiO₂ case after their previous work on anatase TiO₂. This should be addressed more explicitly in both the Abstract and Introduction. In the conclusions, they should also emphasize more clearly what they have learned from their research. Why are the differences in ultrafast electron dynamics crucial for the slower catalytic process? Is the slower speed of CO photooxidation the primary reason for the superior catalytic efficiency of anatase compared to rutile?

The authors claim that the shorter picosecond lifetime of charge carriers in rutile can negatively impact catalytic dynamics on the nanosecond timescale. In my opinion, this is the most significant finding of the study, yet it is somewhat diluted in the text. The authors should reorganize the manuscript to better convey this key result to the reader, particularly in the Discussion and Conclusion sections.

*** Detailed comments ***

:: Introduction (page 3)

- The term "UV illumination" may be misleading for the reader, as the authors use a 770 nm fs-laser pump.

:: Results

- The presence of water in the measurements affects the data and influences the CO oxidation. The authors do not explain why water could not be removed from the sample environment, for example, using a cryo-trap. What was the pressure in the chamber before cooling the samples and injecting the gases?
- What is the pulse intensity of the FEL third harmonic? Do the authors exclude the possibility of undesired space charge effects induced by FEL pulses at the chosen repetition rate?

:: Figure 1

- In Fig. 1a, I suggest adding the intensity scale on the x-axis.
- In Fig. 1a, the black and blue curves are difficult to distinguish.
- In Fig. 1b, I recommend using a logarithmic intensity color scale or adjusting the contrast to better highlight the side bands.

- In Fig. 1b, the black and blue bars are difficult to distinguish.
- In Fig. 1c, I suggest adding the intensity scale on the y-axis.
- In the caption, I recommend defining the binned regions indicated by the bars in panel "b", for example, "blue bar (binding energy region: 460-461 eV)".

:: Figure 2

- It is difficult to assess the quality of the fits. Would it be possible to add error bars to the experimental XPS data?

:: Methods

- The authors state that the sample is tilted by 55 degrees with respect to the fs-laser. However, the angle between the FEL and laser beams is not specified. This is an important parameter, as a large FEL-laser angle could affect the time resolution of the pump-probe scheme.
- The temporal resolution is given as 250 fs, approximately twice the duration of the pump and probe pulses. What is the origin of this value? Is it due to pump-probe jitter, the geometry of the setup, or both?

*** Conclusions ***

The conclusions of the study are well supported by the data analysis and theoretical calculations. The findings are of interest to scientists in the field of chemistry and may also appeal to a broader readership, including researchers in functional materials and ultrafast physics. I recommend publication in Communications Chemistry after revision, as suggested in this report.

Version 1:

Reviewer comments:

Reviewer #1

(Remarks to the Author)

The revised manuscript by Noei et. al. "Dynamics of CO Photooxidation to CO₂ on Rutile (110)" is suitable for publication. They have answered the reviewer comments in a clear and satisfactory way. I look forward to seeing this manuscript in print.

Reviewer #2

(Remarks to the Author)

The authors have address my concerns.

Reviewer #3

(Remarks to the Author)

Thank you for implementing the suggested modifications to the manuscript. It reads much clear and cogent. I can recommend the manuscript to be published in Communications Chemistry.

Reviewer #4

(Remarks to the Author)

The manuscript has been improved by the authors according to the reviewers comments. I recommend publication of the manuscript.

Review manuscript COMMSCHEM-25-0142-T

We thank all reviewers and are grateful to you for the thorough evaluation and constructive feedback. The comments have been highly valuable in improving the clarity and quality of our work. We have addressed all points raised by the reviewers and revised the manuscript accordingly. All changes are marked in the revised version.

Below, we provide a detailed, point-by-point response to each comment.

Best regards,

Heshmat Noei

on behalf of all authors

Comments of Reviewer #1

This manuscript by Gleißner et. al. titled Dynamics of CO photooxidation to CO₂ on Rutile (110) is a study on ultrafast kinetics of CO oxidation monitored by XPS on a FEL. The scope of the manuscript is similar to a previous study presented by the same group on dynamics of CO photooxidation on anatase (101) surfaces using same experimental methodologies. The manuscript is well presented, the conclusions are supported by appropriate experimental observations and the discussions are at appropriate level. I can recommend the manuscript for publication. I have some minor comments and question, mostly to satisfy my own curiosity.

Response: We thank the Reviewer for the positive assessment of our work.

Comment 1: Can the authors comment on why the study was conducted at 80 K instead of at higher temperatures (room temperature for example)? It seems that water adsorption onto the rutile surface (due to cryogenic trapping) is a significant problem. For instance, this study was conducted at 80 K, due to significant water adsorption at 60 K, while analogous experiment reported for anatase was at 60 K. This makes direct comparison of the two data sets more complicated. Wouldn't studies at higher temperature negate this problem?

Response: The temperature of 80 K was chosen to achieve CO adsorption while limiting water adsorption. Pre-experiments on rutile at 60 K led to excessive adsorption of water and CO. This issue was originally discussed on page 15, first line. We have now moved this information to the beginning of the Results section to make it more visible.

Comment 2: In addition, the authors also discussed the potential role of water towards the reaction mechanism. Given the non-observable H₂O signal at the ps regime, what are the implications towards the reaction mechanism?

Response: We thank the Reviewer for bringing up this point. The full spectrum of the O 1s time resolved data for the first 5 min of integration is shown in Fig. S4, indicating that no transient H₂O peak is observed. The presence of a very small OH signal cannot be excluded in the 5 min integrated time resolved data; however, the signal is too small for a quantitative analysis. We cannot rule out that a small amount of OH adsorption may modify the surface electronic structure, which could impact on the reaction dynamics. This topic can be addressed in future investigations. We highlight this point in the discussion on pages 17 and 18 and in the conclusions.

Comment 3: Can the authors comment on the lack of observable signal that can be attributed to physisorbed, chemisorbed molecular O₂ or reduced O₂ on rutile surface (and anatase surface from previous studies)? One of the conclusion of the manuscript is that the fast kinetic observed can be rationalized by a O₂-TiO₂ charge transfer complex. Why this O₂-TiO₂ CT complex is not observable by XPS?

Response: Studies of O₂ adsorption on TiO₂ using XPS are rare. Setvin *et al.* (Ref [44]) observed physisorbed O₂ at 50 K at a binding energy of 537.3 eV in the O 1s spectrum. At 80 K, only a small fraction of O₂ remains adsorbed, and the O₂ signal overlaps with the strong O 1s signal from CO at 536.5 eV, making it impossible to detect it independently in our time-resolved measurements. This limitation has been clarified more explicitly in the revised manuscript (page 11).

Comment 4: For spatial and temporal alignment of the optical laser and FEL pulse, the authors monitor the formation of side bands at 459 eV. In Figure S3, it seems that these side bands occur at t = -0.2 ps delay. Is this because of the relatively large jitter time of the optical laser pulse or an artifact of binning?

Response: The apparent sidebands at t = -0.2 ps is an artifact of the binning. Due to the temporal widths of the optical laser and FEL pulses the sideband are visible at -0.2 and 0.2 ps. This has now been clarified in the Figure caption of Fig. S4 (previously Fig. S3).

Comment 5: Data was collected by raster scanning across the rutile surface to minimize radiation damage (by FEL pulse and also desorption of CO by optical laser pulse). The repetition rate of the FEL pulse is very high at 1 MHz. It seems to me that the sample cannot be feasibly moved to a fresh spot for each shot (speed in the order of 100 m/s). Can the authors comment on how significant is the radiation damage with this current setup and implication to data quality and analysis?

Response: No laser or FEL-induced damage was observed in this experiment. In a previous experiment at FLASH on anatase, laser -induced damage appeared as black spots on the surface. Accordingly, the laser power was adjusted for the current experiment to avoid such effects.

Comments of Reviewer #3

The dynamics of CO oxidation on rutile TiO₂(110) was investigated using free-electron laser X-ray photoemission spectroscopy at 80 K. Building upon a previous study employing the same technique, which observed CO₂ formation on anatase TiO₂(101) between 1.2 and 2.8 ps after time zero, this work demonstrates a similar phenomenon on rutile but on a faster timescale (0 to 0.8 ps). I appreciate the work done by the authors since the experimental results combined with the theory highlights a different reaction mechanism. However, I have following major concerns which I suggest being resolved for making the manuscript more cogent, understandable to a wide range of readers and shed light onto the importance of this work.

Response: We thank the Reviewer for their careful and critical assessment of our work.

Comment 1: The significance and motivation of this research should be more clearly articulated in the introduction. The authors provide an extensive discussion of previously reported ultrafast dynamics of CO oxidation on anatase TiO₂ but then abruptly transition to stating that this study examines the same phenomenon on rutile, with the key distinction being the faster timescale of the oxidation process.

In the introduction, the authors state: *“Different studies [16, 22] found a shorter lifetime of charge carriers in rutile compared to anatase. The reason is that anatase has an indirect band gap that inhibits electron-hole pair recombination, thereby enabling a higher percentage of generated charge carriers to initiate this reaction pathway. The direct band gap of rutile results in a shorter lifetime of charge carriers, thus lowering the catalytic efficiency [16].”* Given that rutile exhibits lower catalytic efficiency than anatase and that ultrafast CO oxidation on anatase has already been reported, what is the broader significance of this study? Beyond the faster timescale of the oxidation process, what additional insights does this work provide? I strongly recommend that the authors revise their introduction to better clarify the significance and impact of their findings.

Response: The word “bulk” was added to the first sentence to emphasize the difference between bulk and surface dynamics. Additionally, the last paragraph of the introduction has been rephrased to address the Reviewer’s comment.

Comment 2: Similarly, both the abstract and introduction lack a statement on the broader applications of this work. I recommend adding a few sentences in both sections to highlight its potential impact.

Response: We thank the Reviewer for this helpful suggestion. We have added statements of the broader applications of this work to the abstract, and the conclusions. They are already given at the beginning of the introduction and we highlighted further potential impact in the “Conclusions” section.

Comment 3: The main text does not provide a detailed description of Figure 1. For instance: (i) What do the red, blue, and black line profiles represent in Figure 1a? (ii) What do the pink and yellow shaded regions indicate in Figure 1c? (iii) Figure 1b is not mentioned in the text at all. Since the primary results are presented from Figure 2 onward and Figure 1 primarily illustrates prerequisite details of the experimental setup, it may be appropriate to move

Figure 1 to the Supplementary. However, if the authors choose to retain it in the main text, it should be described more thoroughly.

Response: Figure 1 has been moved to the Supplementary Information and the color code is now explained, as suggested by the Reviewer.

Comment 4: The data presented in Figure 2, the rise and fall of CO₂ concentration over time appear asymmetric, with the decline being longer than the rise. Is this asymmetry an experimental artifact, or does it reflect a genuine physical phenomenon?

Response: Considering the experimental error bars, we believe that this slight asymmetry should not be overinterpreted. It likely does not indicate a significant physical effect but rather falls within the expected experimental uncertainty.

Comment 5: Does the 800-fs time window for CO₂ formation remain consistent regardless of the flash-annealing duration? Specifically, if the same data presented in Figure 2 were plotted after flash-annealing the surface for more than 30 minutes, would the time window still be 0 to 800 fs?

Response: We cannot directly address this question, as the annealing time was always kept constant (below 1 min). The procedure involved only flash-annealing without maintaining the sample at elevated temperature for longer periods. This information is now has been added to the "Methods" section of the manuscript.

Comment 6: I appreciate the detailed "Discussion" section. However, the reaction mechanism would be more accessible to a general reader if the authors included a schematic representation. For instance, in the discussion, the authors state: "*The CO₂ signal therefore implies that oxygen was reduced, dissociated, and reacted with CO to form CO₂ within 800 fs.*" I recommend illustrating this process through a schematic, like the one presented in the abstract of reference 24.

Response: We thank the Reviewer for this helpful suggestion. A schematic illustrating the reaction mechanism has now been added to the "Discussion" section as Fig. 4.

Comment 7: The reason for the difference in CO oxidation dynamics between rutile and anatase is not clearly stated in the manuscript. While the authors mention several possible factors, such as temperature differences and the presence of water, a concise statement summarizing the key factor driving this difference should be included at the end of the discussion section or in the conclusion.

Response: We agree with the Reviewer and have re-organized the discussion section. Based on our data and calculations, we cannot not be able to give the final answer to this question. Further time-resolved calculations would help to answer this question, we point this now out in the "Conclusions" section.

Comments of Reviewer #4

Report on the manuscript entitled "Dynamics of CO Photooxidation to CO₂ on Rutile (110)" by Helena Gleißner et al.

*** General comments ***

Comment 1: The manuscript presents a very detailed study on the ultrafast photoinduced oxidation dynamics of CO to CO₂ on the surface of rutile TiO₂ catalysts. This work is essentially a continuation of the authors' previous research on the same mechanism in anatase TiO₂, which was published in the literature a few years ago. This type of investigation, which combines both experimental and theoretical innovative methods, is highly timely and has a significant impact on various fields, including research on sustainable energy sources. I enjoyed reading the manuscript; however, at times, I had the impression that some sections are somewhat redundant, and the level of detail is excessive. I encourage the authors to revise the text, moving non-essential parts into the Methods and Theoretical sections. This would enhance readability and improve the overall flow of the manuscript.

More importantly, the authors should better articulate their motivation for investigating the rutile TiO₂ case after their previous work on anatase TiO₂. This should be addressed more explicitly in both the Abstract and Introduction.

Response: We thank the Reviewer for the positive assessment of our work. We streamlined the introduction and revised the discussion section accordingly.

Comment 2: In the conclusions, they should also emphasize more clearly what they have learned from their research. Why are the differences in ultrafast electron dynamics crucial for the slower catalytic process? Is the slower speed of CO photooxidation the primary reason for the superior catalytic efficiency of anatase compared to rutile?

The authors claim that the shorter picosecond lifetime of charge carriers in rutile can negatively impact catalytic dynamics on the nanosecond timescale. In my opinion, this is the most significant finding of the study, yet it is somewhat diluted in the text. The authors should reorganize the manuscript to better convey this key result to the reader, particularly in the Discussion and Conclusion sections.

Response: We have revised the Conclusions and Discussion section to more clearly convey the key findings of our study.

*** Detailed comments ***

:: Introduction (page 3)

Comment 3: The term "UV illumination" may be misleading for the reader, as the authors use a 770 nm fs-laser pump.

Response: The paragraph was edited for clarification.

:: Results

Comment 4: The presence of water in the measurements affects the data and influences the CO oxidation. The authors do not explain why water could not be removed from the sample environment, for example, using a cryo-trap. What was the pressure in the chamber before cooling the samples and injecting the gases?

Response: A cryo trap was not available in this experimental setup. The residual pressure was $4e-10$ mbar in the main chamber before dosing CO and O₂. We have included this information now on page 9 of the manuscript.

Comment 5: What is the pulse intensity of the FEL third harmonic? Do the authors exclude the possibility of undesired space charge effects induced by FEL pulses at the chosen repetition rate?

Response: The pulse energy of the third harmonic was 25-30 μ J. No space charge effects (in form of shifts in the spectra) were observed.

:: Figure 1

Comment 6: - In Fig. 1a, I suggest adding the intensity scale on the x-axis.

- In Fig. 1a, the black and blue curves are difficult to distinguish.

- In Fig. 1b, I recommend using a logarithmic intensity color scale or adjusting the contrast to better highlight the side bands.

- In Fig. 1b, the black and blue bars are difficult to distinguish.

- In Fig. 1c, I suggest adding the intensity scale on the y-axis.

- In the caption, I recommend defining the binned regions indicated by the bars in panel "b", for example, "blue bar (binding energy region: 460-461 eV)".

Response: The suggestions were incorporated into the Figure, now Fig. S1.

:: Figure 2

Comment 7: It is difficult to assess the quality of the fits. Would it be possible to add error bars to the experimental XPS data?

Response: The error bars of the rel. CO₂ amount are given. The error bars depend on the CO and CO₂ fit of each spectrum. Information about the fitting is given at the end of the experimental part of the "Methods" section.

:: Methods

- The authors state that the sample is tilted by 55 degrees with respect to the fs-laser. However, the angle between the FEL and laser beams is not specified. This is an important parameter, as a large FEL-laser angle could affect the time resolution of the pump-probe scheme.

Response: The laser beam and FEL are colinear. The information is now added to the Experimental section.

- The temporal resolution is given as 250 fs, approximately twice the duration of the pump and probe pulses. What is the origin of this value? Is it due to pump-probe jitter, the geometry of the setup, or both?

Response: We thank the reviewer for the question. The temporal resolution was experimentally determined from the width of the sidebands. It combined contributions of the Laser pulses, FEL pulse duration and the x-ray optics.

*** Conclusions ***

The conclusions of the study are well supported by the data analysis and theoretical calculations.

The findings are of interest to scientists in the field of chemistry and may also appeal to a broader readership, including researchers in functional materials and ultrafast physics.

I recommend publication in Communications Chemistry after revision, as suggested in this report.

The dynamics of CO oxidation on rutile TiO₂(110) was investigated using free-electron laser X-ray photoemission spectroscopy at 80 K. Building upon a previous study employing the same technique, which observed CO₂ formation on anatase TiO₂(101) between 1.2 and 2.8 ps after time zero, this work demonstrates a similar phenomenon on rutile but on a faster timescale (0 to 0.8 ps). I appreciate the work done by the authors since the experimental results combined with the theory highlights a different reaction mechanism. However, I have following major concerns which I suggest being resolved for making the manuscript more cogent, understandable to a wide range of readers and shed light onto the importance of this work.

1. The significance and motivation of this research should be more clearly articulated in the introduction. The authors provide an extensive discussion of previously reported ultrafast dynamics of CO oxidation on anatase TiO₂ but then abruptly transition to stating that this study examines the same phenomenon on rutile, with the key distinction being the faster timescale of the oxidation process.

In the introduction, the authors state: *“Different studies [16, 22] found a shorter lifetime of charge carriers in rutile compared to anatase. The reason is that anatase has an indirect band gap that inhibits electron-hole pair recombination, thereby enabling a higher percentage of generated charge carriers to initiate this reaction pathway. The direct band gap of rutile results in a shorter lifetime of charge carriers, thus lowering the catalytic efficiency [16].”* Given that rutile exhibits lower catalytic efficiency than anatase and that ultrafast CO oxidation on anatase has already been reported, what is the broader significance of this study? Beyond the faster timescale of the oxidation process, what additional insights does this work provide? I strongly recommend that the authors revise their introduction to better clarify the significance and impact of their findings.

2. Similarly, both the abstract and introduction lack a statement on the broader applications of this work. I recommend adding a few sentences in both sections to highlight its potential impact.

3. The main text does not provide a detailed description of Figure 1. For instance: (i) What do the red, blue, and black line profiles represent in Figure 1a? (ii) What do the pink and yellow shaded regions indicate in Figure 1c? (iii) Figure 1b is not mentioned in the text at all.

Since the primary results are presented from Figure 2 onward and Figure 1 primarily illustrates prerequisite details of the experimental setup, it may be appropriate to move Figure 1 to the Supplementary. However, if the authors choose to retain it in the main text, it should be described more thoroughly.

4. The data presented in Figure 2, the rise and fall of CO₂ concentration over time appear asymmetric, with the decline being longer than the rise. Is this asymmetry an experimental artifact, or does it reflect a genuine physical phenomenon?
5. Does the 800-fs time window for CO₂ formation remain consistent regardless of the flash-annealing duration? Specifically, if the same data presented in Figure 2 were plotted after flash-annealing the surface for more than 30 minutes, would the time window still be 0 to 800 fs?
6. I appreciate the detailed “Discussion” section. However, the reaction mechanism would be more accessible to a general reader if the authors included a schematic representation. For instance, in the discussion, the authors state: *“The CO₂ signal therefore implies that oxygen was reduced, dissociated, and reacted with CO to form CO₂ within 800 fs.”* I recommend illustrating this process through a schematic, like the one presented in the abstract of reference 24.
7. The reason for the difference in CO oxidation dynamics between rutile and anatase is not clearly stated in the manuscript. While the authors mention several possible factors, such as temperature differences and the presence of water, a concise statement summarizing the key factor driving this difference should be included at the end of the discussion section or in the conclusion.